# Comparative Bone-Protective Effects of Tocotrienol Isomers from Palm and Annatto in Dexamethasone-Induced Osteoporotic Male Rats

**DOI:** 10.3390/ijms262010206

**Published:** 2025-10-20

**Authors:** Elvy Suhana Mohd Ramli, Fairus Ahmad, Nur Aqilah Kamaruddin, Kok-Yong Chin, Ima Nirwana Soelaiman, Kok-Lun Pang

**Affiliations:** 1Department of Anatomy, Faculty of Medicine, Universiti Kebangsaan Malaysia, Jalan Yaacob Latif, Cheras, Kuala Lumpur 56000, Malaysia; fairusahmad@ukm.edu.my (F.A.); nur.aqilah@ukm.edu.my (N.A.K.); 2Department of Pharmacology, Faculty of Medicine, Universiti Kebangsaan Malaysia, Jalan Yaacob Latif, Cheras, Kuala Lumpur 56000, Malaysia; chinky@ukm.edu.my (K.-Y.C.); imasoel@ppukm.ukm.edu.my (I.N.S.); koklun.pang@monash.edu (K.-L.P.); 3Jeffrey Cheah School of Medicine and Health Sciences, Monash University Malaysia, Bandar Sunway, Subang Jaya 46150, Malaysia

**Keywords:** tocotrienol, bone metabolism, dexamethasone-induced osteoporosis, oxidative stress

## Abstract

Oxidative stress brought on by prolonged glucocorticoid therapy reduces bone growth, structure, and mechanical qualities. Free radicals promote osteoclastic activity and are harmful to osteoblasts. As an antioxidant, tocotrienol offers protection against illnesses linked to free radicals. Annatto tocotrienol (ATT) is a tocopherol-free tocotrienol, and palm tocotrienol (PTT) is a tocotrienol mixture. Finding out how ATT and PTT protect against glucocorticoid-induced osteoporosis was the aim of this study. In this study, 32 mature male Sprague-Dawley rats were employed. Twenty-four rats were divided into three groups: Dex, Dex + PTT, and Dex + ATT, after being adrenalectomized. A sham surgery was performed on the remaining eight rats. The Dex group received oral vehicle palm olein (0.1 mL/kg/day) and intramuscular injection of dexamethasone (120 µg/kg/day). Dexamethasone 120 µg/kg/day was administered intramuscularly to the Dex + PTT and Dex + ATT group, and palm tocotrienol (PTT) and annatto tocotrienol (ATT) 60 mg/kg/day were added as a supplement. Vehicle palm olein was administered intramuscularly to the sham-operated rats, 0.05 mL/kg/day and 0.1 mL/kg/day orally. The treatments were administered for two months before the rats were euthanized. The biomechanical strength of the femoral bones was evaluated, and the structural characteristics of bone histomorphometry were examined. According to the findings, prolonged glucocorticoid therapy resulted in decreased superoxide dismutase (SOD) activity, increased lipid peroxidation, and bone carboxy-terminal collagen cross-linkages (CTX). Bone Volume/Tissue Volume (BV/TV) and Trabecular Number (Tb.N) were drastically reduced, which severely reduced bone biomechanical strength. There were also alterations in the bone formation and resorption gene expressions. Lipid peroxidation, CTX levels, and SOD activity were all considerably maintained at control levels by PTT and ATT intake. Additionally, it preserved the biomechanical strength and bone structure, as well as maintaining the gene expressions. According to the study’s findings, ATT and PTT may have anabolic and anti-resorptive properties and have the potential to be utilized as a prophylactic for individuals receiving long-term glucocorticoid therapy.

## 1. Introduction

Glucocorticoids impair bone remodeling by reducing formation and increasing resorption, leading to weaker bones and higher fracture risk, through direct effects on bone cells or indirect actions via extra-skeletal tissues [1,2]. Glucocorticoids cause bone loss by increasing osteoclast activity and reducing bone formation. They block osteoblast growth, differentiation, and survival, lower collagen and matrix production, and trigger apoptosis of osteoblasts and osteocytes. As a result, less bones are formed during each remodeling cycle [3]. The primary mechanism underlying chronic glucocorticoid-induced bone loss is the diminished osteoblastic lineage, which leads to decreased bone production [4].

Glucocorticoids have inconsistent effects on bone resorption. Glucocorticoids indirectly increase the lifespan and activity of osteoclasts, enhancing bone resorption [5,6]. According to certain human research, glucocorticoids increase resorption parameters [7]. Dexamethasone promotes the growth of osteoclast-like cells [8]. Glucocorticoids increase the expression of the receptor activator of NF-kappa B ligand (RANKL) and decrease the expression of its soluble decoy receptor, osteoprotegerin (OPG), in stromal and osteoblastic cells, which contribute to the increase in bone resorption [9,10]. Dexamethasone (Dex) was also found to stimulate osteoclast-like cell formation [8].

Chronic glucocorticoid use increases oxidative stress [11]. Oxidized lipids and free radicals promote osteoclast activity and block osteoblast maturation, leading to greater bone resorption [12,13]. During osteoclast activation by parathyroid hormone (PTH), interleukin -11 (IL-1), and tumour necrosis factor alpha (TNF), superoxide anions play a key role in bone breakdown. Osteoclasts contain superoxide dismutase (SOD) and produce free radicals when stimulated by these hormones [14].

Tocotrienols, a form of vitamin E, protect bone by being absorbed more effectively into cells. They enhance bone mineralization, stimulate osteoblast differentiation, and reduce osteoclast formation, supporting overall bone health [15]. Vitamin E prevents lipid peroxidation. Its forms include four tocopherols (α, β, γ, δ) and four similar tocotrienols. Besides being antioxidants, they also show anti-cancer, neuroprotective, antiplatelet, and cholesterol-lowering effects. Notably, α-tocotrienol has 40–60 times stronger antioxidant activity against lipid peroxidation than α-tocopherol [16]. Animal studies show that tocopherols and tocotrienols help preserve bone density, prevent further loss, and support trabecular bone formation. A mix of tocotrienols is also more effective than α-tocopherol in protecting trabecular bone from free radical damage [17,18].

Palm tocotrienols (PTT), a form of vitamin E from palm oil, are studied for their antioxidant, anti-inflammatory, and bone-protective effects. They reduce oxidative stress, block cytokines like TNF-α and interleukin 6 (IL-6) that drive bone loss, boost osteoblast activity, and suppress osteoclasts, helping maintain bone balance. Rat studies show that tocotrienol supplements improve bone density, strength, and structure, especially in osteoporosis models [19]. Early clinical trials suggest tocotrienols may support bone health in postmenopausal women and the elderly, though larger studies are needed [20]. They protect bone better than tocopherols because of stronger antioxidant effects and higher cellular uptake [21].

Annatto tocotrienol (ATT), a natural vitamin E from the seeds of the South American *Bixa orellana* plant, contains only tocotrienols—mainly δ-tocotrienol (≈90%) and γ-tocotrienol (≈10%)—without tocopherols, which can block their activity. It has strong antioxidant, anti-inflammatory, and cholesterol-lowering effects. In estrogen-deficient rats, it boosted the bone formation marker osteocalcin, lowered the resorption marker CTX, and improved bone structure. Combined with lovastatin, it also showed bone-building effects [22,23].

There is a need for safe and effective treatments to prevent steroid-induced osteoporosis and fractures in patients with Cushing’s syndrome or long-term glucocorticoid use. Current therapies improve bone density but have a limited impact on fracturing risk. Palm and annatto tocotrienols show promising benefits for bone health [17,23].

In Sprague-Dawley rats, bone mass levels remain stable between 3 and 9 months, when they are considered young adults [24]. To maintain high glucocorticoid levels, the adrenal glands were removed, and dexamethasone, a synthetic glucocorticoid, was given instead. The treatment dose and duration were based on a previous study [25,26]. Since mineralocorticoids have little effect on bone, no replacement was given [27]. To maintain salt balance, the animals received regular saline solutions to drink. The dose of annatto tocotrienol was based on earlier studies [19].

The present study aimed to compare the bone-protective effects of tocotrienol isomers derived from palm and annatto in dexamethasone-induced osteoporotic male rats, with specific evaluation of their impact on bone strength, microarchitecture, turnover markers, oxidative stress, and histomorphometric parameters.

## 2. Results

### 2.1. Bone Biochemical Markers

The osteocalcin level showed no significant changes after two months of dexamethasone treatment in the Dex group compared to the Sham and Baseline groups. Similarly, no significant changes were observed in the Dex + PTT group as well as the Dex + ATT group (Figure 1a).

CTX levels were significantly elevated in all the groups compared to the pre-treatment level. CTX levels in dexamethasone-treated adrenalectomized rats (Dex) were significantly higher than those in the Sham group. Rats supplemented with PTT and ATT had significantly lower post-treatment CTX levels, compared to the Dex group. Notably, there was no significant difference in CTX levels between the groups receiving the PTT and ATT compared to the Sham group (Figure 1b).

### 2.2. Effects of Annatto Tocotrienol on Bone Structural Histomorphometry in Dexamethasone-Induced Osteoporotic Bone

The Bone Volume/Tissue Volume (BV/TV) (Figure 2a), Trabecular Number (Tb.N) (Figure 2b) and Trabecular Thickness (Tb.Th) (Figure 2c) were significantly reduced after 2 months in Dex group compared to Sham group, but there was no significant changes in the Trabecular Separation (Tb.Sp) (Figure 2d). The structural histomorphometric parameter was preserved when PTT and ATT 60 mg/kg/day were added to the Dex group. Comparing the Dex + ATT group to the Dex group, the Tb.N and BV/TV were significantly higher, and the Tb.Sp was significantly lower. Indicating an increase in bone mass, the BV/TV, Tb.N, and Tb.Sp of both the PTT and ATT supplemented groups did not differ significantly from those of the Sham group. However, the Tb.Th. did not significantly change as a result of PTT and ATT intake. The photomicrograph in Figure 3 shows the alterations.

### 2.3. Static Parameters

In comparison to the Sham group, the Dex group showed a substantial drop in osteoblast surface (Ob.S/BS) after two months of dexamethasone treatment, but no significant change in osteoclast surface (Oc.S/BS). The osteoid volume (OV/BV) and (OS/BS) were also significantly lower than the Dex group. Supplementing with PTT and PTT had maintained the Ob.S/BS while drastically lowering the Oc.S/BS. Dex + ATT and Dex + PTT groups’ Ob.S/BS and Oc.S/BS did not differ significantly from the Sham group (Figure 4a,b). Similar results were observed in the OS/BS and OV/BV (Figure 4c,d). The changes are illustrated in Figure 5. The Dex group showed a marked reduction in the trabecular bone, which was thinner and more separated compared to the Sham group. Conversely, rats supplemented with tocotrienol (Dex + ATT and Dex + PTT) demonstrated more, thicker, and less separated trabecular compared to the Dex group.

### 2.4. Dynamic Parameters

Dynamic parameter analysis revealed that the Dex group’s mineralized surface (MS/BS), mineral apposition rate (MAR), and bone formation rate (BFR) were all considerably lower after two months of dexamethasone treatment than the Sham group. The MS/BS, MAR, and BFR Dex + PTT and Dex + ATT groups were significantly maintained by palm tocotrienol and annatto tocotrienol administration, which reversed the effects of dexamethasone. The MS/BS, MAR, and BFR of the Dex + PTT and Dex + ATT groups did not change significantly from those of the Sham group (Figure 6). The changes in the dynamic parameter are shown in the photomicrographs in Figure 7.

### 2.5. Effects of Tocotrienols on Bone Biomechanical Strength in Dexamethasone-Induced Osteoporotic Bone

Both intrinsic (Young’s modulus, stress, and strain) and extrinsic (energy, load, and flexure extension) parameters of the femoral bone were significantly lower in the Dex rats after two months of dexamethasone administration (Figure 8a–c) compared to the Sham group. PTT and ATT prevented the changes to some of the extrinsic and intrinsic qualities of the bones (Figure 9a–c). PTT showed superior effects compared to ATT as it maintained all parameters except for maximum extension, whereas ATT only preserved the stress, energy, and maximum load.

### 2.6. Effects of Tocotrienols on Bone Formation and Resorption Gene Expressions in Dexamethasone-Induced Osteoporotic Bone

The expression of the genes related to bone formation in the Dex rats differed significantly from that of the Sham rats. In the Dex group, the expression of the integrin-binding sialoprotein (IBSP) and osterix (SP7) genes was significantly lower (*p* < 0.05), whereas the expression of the osteocalcin and COL1α1 genes increased dramatically. The Dex group’s RANKL gene expression did not differ significantly from that of the Sham group, but the osteoprotegrin gene’s expression was significantly lower, which resulted in a considerable rise in the RANKL/OPG ratio. The expression of additional genes linked to bone resorption, such as MMP-9, osteopontin, and cathepsin K, also significantly increased. The effect of dexamethasone on genes associated with bone formation was considerably reversed by treatment with PTT and ATT; osteocalcin and COL1α1 were reduced, while the expression of the osterix gene showed significant elevation (Figure 10). In comparison to the Dex group, the IBSP gene’s expression was not significantly altered by PTT and ATT administration (Figure 11).

### 2.7. Effects of Tocotrienols on Lipid Peroxidation and Oxidative Stress Enzymes in Dexamethasone-Induced Osteoporotic Bone

Superoxide dismutase (SOD) activity was markedly reduced (Figure 12b), while glutathione peroxidase (GPX) activity was elevated in the femoral bone following two months of dexamethasone administration in adrenalectomized rats (Figure 12c). Catalase (CAT) activity, however, remained unchanged (Figure 12d). Malondialdehyde (MDA), an indicator of lipid peroxidation, was significantly increased in the Dex group, reflecting enhanced oxidative stress (Figure 12a). In contrast, Dex rats supplemented with ATT or PTT demonstrated significantly higher SOD activity and lower GPX activity compared to the Dex group, with levels comparable to those of the Sham group. Tocotrienol supplementation did not significantly affect CAT activity but significantly reduced MDA levels relative to the Dex group (Figure 12a–d).

## 3. Discussion

Two months of dexamethasone treatment caused oxidative stress in bone, shown by higher MDA levels (lipid peroxidation). It lowered SOD activity, left CAT unchanged, and increased GPX activity. Overall, this shows that prolonged dexamethasone disrupts the balance between ROS production and antioxidant defenses [28]. Both clinical and experimental evidence has demonstrated that oxidative stress plays a critical role in the pathogenesis of osteoporosis [29]. Increased oxidative stress impairs skeletal health by suppressing osteoblast differentiation and survival while promoting osteoclast differentiation and bone resorption [30,31]. In this study, Dex rats showed higher MDA levels, indicating more lipid peroxidation, a process linked to osteoporosis. Parhami et al. (1997) also reported that osteoporotic bone accumulates lipids that easily oxidize, worsening bone damage [31]. Oxidized lipids promote bone resorption by stimulating osteoclasts. The drop in SOD activity may have caused superoxide buildup, while high lipid peroxidation and excess superoxide increased oxidative stress, making bone more vulnerable to injury. Antioxidant enzymes help protect against this, with GPX especially important in reducing lipid hydroperoxides and limiting further damage [32]. The increase in GPX activity after long-term dexamethasone treatment may be a compensatory response to higher superoxide levels caused by reduced SOD activity. GPX helps neutralize superoxide and hydrogen peroxide by converting them into water. However, earlier studies found that glucocorticoids lowered GPX levels in hippocampal cell cultures, suggesting that the antioxidant response may vary between tissues [33].

Bone turnover markers show how active bone remodeling is. Formation markers (like osteocalcin, ALP, and P1NP) come from osteoblasts during bone building, while resorption markers (like PYD, DYP, and CTX-1) reflect bone breakdown. In this study, although osteoclast numbers did not increase significantly, higher lipid peroxidation and lower SOD activity indicated more free radicals, which may have promoted osteoclast activity. The rise in CTX levels supports increased bone resorption. Osteoclasts can also release free radicals at resorption sites, further speeding up bone breakdown [34]. Moreover, glucocorticoids are known to promote osteoblast apoptosis, extend osteoclast lifespan, and disrupt the coupling between these cells [35]. The sharp drop in Ob.S/BS seen in this study suggests that oxidative stress reduced osteoblast activity and bone formation. However, tocotrienol did not significantly affect osteocalcin levels, matching earlier reports where bone formation markers showed little or no change [36,37,38]. Nevertheless, osteocalcin levels did not fully reflect the decline in bone formation. Hoola et al. (2019) reported a weak negative correlation between serum osteocalcin levels and glucocorticoid dosage, suggesting that osteocalcin may not be a sensitive marker of glucocorticoid-induced changes in bone metabolism [39]. Furthermore, osteocalcin levels are known to decrease with age, which may confound its reliability as an indicator of bone formation in this context.

Bone histomorphometry can directly assess bone microarchitecture, remodeling, and cell activity, unlike densitometry or serum markers. After two months of dexamethasone treatment, bone volume and structure worsened, with significant declines in BV/TV, Tb.N, and Tb.Th. These changes likely result from oxidative stress, which weakens osteoblast function and enhances osteoclast activity. Fewer, less active osteoblasts reduce matrix production, lowering wall thickness, trabecular volume, and thickness. Since osteoblasts control bone formation in each remodeling cycle, these impairments reduce overall bone replacement [40]. Consistent with these findings, Chavassieux et al. (2000) reported significantly reduced mineralizing surfaces and bone volume in glucocorticoid-treated patients compared with controls [41]. In this study, dexamethasone treatment led to higher Oc.S/BS, Ov/BV, and Os/BS, while Ob.S/BS decreased. Lower MS/BS, MAR, and BFR confirmed reduced osteoblast activity, matching earlier findings [42]. Evidence shows that glucocorticoids cause early, rapid bone loss mainly by directly stimulating osteoclasts. However, even when osteoclast expansion is blocked, glucocorticoids still trigger osteocyte death and impair osteoblast function. By extending osteoclast lifespan, glucocorticoids cause an early but temporary increase in bone remodeling [43].

The differentiation and functional activity of osteoblasts and osteoclasts are regulated by a complex cascade of genes [44]. In this study, dexamethasone reduced the expression of osteoblast-related genes like Sp7 and IBSP, but increased Col1α1 and Bglap. The drop in osteoblast-specific genes may be due to oxidative stress, which causes osteoblast and osteocyte death, lowering the number of mature osteoblasts [45,46]. Clinically, patients on dexamethasone show reduced serum osteocalcin and type I procollagen levels, indicating poor bone formation [47]. In this study, the rise in Col1α1 and osteocalcin genes may be a compensatory response to increased bone resorption. Kim et al. (2006) also noted that osteoclast activity can trigger new bone formation as part of the remodeling process [48].

This study showed a marked increase in osteoclast-related genes, including Spp1, Itgb3, and Ctsk. Although osteoclast numbers did not rise significantly, the results suggest that high glucocorticoid levels may promote osteoclast activity through oxidative stress. Jia et al. (2011) also found that glucocorticoids boost Ctsk expression, which lowers collagen in resorption sites and contributes to reduced BV/TV [49,50]. Dexamethasone also increased Itgb3 expression and binding [51]. Osteopontin helps start bone resorption by anchoring osteoclasts to bone, and higher levels are linked to faster bone turnover and lower bone mineral density (BMD) [52,53]. In this study, RANKL expression was unchanged, but OPG was significantly reduced, raising the RANKL/OPG ratio [54]. This imbalance likely promoted osteoclast activity. Previous studies show glucocorticoids raise the RANKL/OPG ratio by boosting RANKL and lowering OPG, driving bone loss [55,56]. Since OPG also promotes osteoclast apoptosis, its reduction may prolong osteoclast survival and activity [57].

Our study found that reduced bone strength was linked to structural damage. Both extrinsic properties (energy, load, maximum flexure extension) and intrinsic values (Young’s modulus, stress, strain) were decreased. These results match earlier histomorphometric findings in glucocorticoid-treated patients [10,58]. Previous research shows that glucocorticoids reduce bone strength by 20% with trabecular thinning, 70% with trabecular loss, and 77% with both thinning and loss [59]. High doses of glucocorticoids cause major changes in bone quantity, shape, and formation, especially in the axial skeleton [46]. They disrupt the osteocyte–canalicular network, increasing dead osteocytes and empty lacunae, which reduces the bone’s ability to repair microdamage and weakens its strength [60]. Since remodeling happens only on bone surfaces, trabecular bone responds faster than other bone types to changes in bone balance [61].

Tocotrienol supplements helped protect bone strength in rats. With supplementation, femoral bone stress (intrinsic) and energy, maximum extension, and maximum load (extrinsic) did not decrease. However, annatto tocotrienol could not prevent dexamethasone from affecting other properties like strain and Young’s modulus, while palm tocotrienol showed stronger protective effects. Studies by Shuid et al. (2008, 2010) reported similar findings [62,63]. Both palm and annatto tocotrienols reduced dexamethasone-induced bone resorption, as shown by stable CTX levels, helping preserve bone structure. Histomorphometric analysis confirmed that both forms restored bone volume (BV/TV), trabecular number (Tb.N), and spacing (Tb.Sp). They also maintained femur structure in testosterone-deficient rats [64].

Antioxidants and related enzymes protect against oxidative damage. Tocotrienols, with strong antioxidant properties, may help prevent membrane lipid peroxidation [40]. In this study, we used palm tocotrienol and annatto tocotrienol (90% delta, 10% gamma), the only tocopherol-free source. Annatto tocotrienol may offer a new way to prevent glucocorticoid-induced osteoporosis.

Palm and annatto tocotrienol supplements prevented lipid peroxidation and preserved bone SOD activity, though annatto did not significantly affect GPX or CAT. Similar studies showed palm tocotrienol lowers lipid peroxidation in both normal and ovariectomized rats [65,66]. Annatto’s antioxidant effects may have reduced oxidative stress, limiting osteoclast activity and bone resorption, as shown by lower CTX levels. Both tocotrienols also preserved osteoblast surface and reduced osteoclast surface, explaining the drop in CTX. However, osteocalcin levels did not match osteoblast changes, possibly due to natural daily fluctuations, which remain poorly understood [67].

A recent study showed that annatto tocotrienol blocked free radical–induced osteoclast growth and reduced bone resorption in testosterone-deficient rats [68]. It may protect osteoblasts from oxidative damage, helping maintain their number, activity, and bone formation, which supports bone strength and structure. Like palm tocotrienol, annatto did not significantly change GPX activity [37], though palm tocotrienol has been shown to increase GPX in ovariectomized rats [69]. In this study, annatto tocotrienol lowered oxidative stress in bones after dexamethasone treatment [70], likely protecting against glucocorticoid damage. Similar results were reported by Maniam et al. (2008) [71].

Fujita et al. (2012) reported that vitamin E did not affect osteoblast growth but increased bone resorption and reduced bone mass by promoting osteoclast fusion [72]. Earlier studies also found a negative link between alpha-tocopherol and bone mass. In contrast, many studies show that tocotrienol protects against bone loss in various osteoporotic rat models. Comparisons of tocopherol and tocotrienol found that tocotrienol offered better protection based on cytokines, biomarkers, calcium, and bone structure [72,73,74]. However, in some cases, alpha-tocopherol was as effective or even better than tocotrienol [64]. In this study, we used a tocopherol-free tocotrienol mix (annatto tocotrienol).

Our findings showed that PTT and ATT increased Sp7 gene expression but lowered osteocalcin and Col1a1. They also reduced osteoclast-related genes (ctsk, itgb3, spp1) without affecting the RANKL/OPG ratio, suggesting an anti-osteoclast effect. Cathepsin K inhibitors are known treatments for osteoporosis by blocking bone resorption [64]. Similarly, blocking αvβ3 integrin reduces resorption without lowering osteoclast numbers [75]. Osteopontin not only aids osteoclast recruitment but also influences crystal formation [76]. Thus, palm tocotrienol’s ability to suppress osteopontin, αvβ3 integrin, and cathepsin K highlights its potential as an osteoporosis therapy.

Osterix (OSX) activates the Col1a1 promoter and is essential for osteoblast differentiation [77]. The increase in OSX expression suggests that tocotrienol may enhance osteoblast differentiation and bone formation. However, both ATT and PTT also reduced osteoclast activity and bone resorption, which can indirectly limit new bone formation, as shown by the lower expression of osteoclast-related genes. Other studies support these findings. For example, Abu Khatir et al. reported that palm vitamin E supplementation boosted Runx2, Osterix, and BMP-2 expression in a nicotine-induced osteopenia model [78]. Similarly, Chin and Ima-Nirwana showed that annatto tocotrienol could upregulate bone formation genes such as alkaline phosphatase, β-catenin, collagen type I alpha 1, and osteopontin.

Fracture prevention in glucocorticoid-induced osteoporosis is multifactorial. Although annatto tocotrienol improved bone structure, this did not translate into greater bone strength, indicating that structural improvement alone may not account for fracture risk. In contrast, palm tocotrienol exhibited better protection of bone strength, although with similar bone formation effects. Glucocorticoids primarily impair bone by suppressing bone formation and inducing osteocyte death, whereas both palm and annatto tocotrienols mainly act by reducing bone resorption. Evidence from the GPRD further shows that glucocorticoid users face a higher fracture risk than non-users even at the same BMD, suggesting that fracture risk cannot be fully explained by BMD alone. The tested dose of the tocotrienols lowered bone resorption but may not have been enough to increase bone formation. More studies with higher doses or longer treatment are needed.

In this study, only a single dose of PTT and ATT (60 mg/kg/day) were tested. Its effects at the higher or lower doses were not determined. Radiological examination, such as microCT which can provide better bone structural evaluation, was not performed in this study but will be considered in the future. There are also other limitations, as while gene expression analysis was performed, the corresponding protein expression levels were not assessed. Confirming gene expression data with protein-level validation is essential to ensure that transcriptional changes translate into functional protein alterations. Additionally, although histomorphometric analysis was conducted, the use of micro-computed tomography (microCT) could have provided a more precise and quantitative assessment of bone structural parameters, particularly in differentiating cortical and trabecular bone compartments. Although the current study has suggested mechanism reflected by changes in osteoblast and osteoclast related gene changes with tocotrienol treatment (Figure 13), it did not evaluate key signaling pathways, such as the Wnt/β-catenin pathway or sclerostin expression, which play critical roles in mediating the effects of glucocorticoids on bone formation and resorption. Inclusion of these analyses would offer deeper insight into the molecular mechanisms underlying glucocorticoid-induced osteoporosis.

## 4. Materials and Methods

### 4.1. Animals and Treatment

This is an in vivo study using an animal model of glucocorticoid-induced osteoporosis. All procedures were carried out in accordance with the institutional guidelines for animal research and approved by the Universiti Kebangsaan Malaysia UKM Research and Animal Ethics Committee. Using male rats provides a more stable hormonal baseline, allowing for clearer evaluation of the interventions and their effects on bone microarchitecture, cellular activity, and oxidative stress biomarkers.

Thirty-two male Sprague–Dawley rats, aged 3 months, weighing between 280 and 300 g, were acquired from the Animal Breeding Center of Universiti Kebangsaan Malaysia. Eight rats underwent a sham procedure, and twenty-four rats were adrenalectomized. A 1:1 mixture of Ketapex and Xylazil (Troy Laboratories, Glendenning, Australia) at a dose of 0.1 mL/kg was used to anesthetize the rats. Bilateral flank muscle incisions and the dorsal midline skin were made in order to view the adrenal glands. Prior to removal, the adrenal glands were located, and the arteries were ligated to stop the bleeding. Normal saline was used to clean and repair the wounds.

The adrenalectomized rats were divided randomly into three groups of 8, and another group of 8 was the sham-operated rats which made up 4 groups of animals. Each group consisted of 8 animals. Two weeks following the adrenalectomy, the corresponding treatments were initiated. The following treatments were administered to the rats: Sham: sham-operated group, given vehicle palm olein 0.05 mL/100 g by intramuscular (IM) injection and 0.1 mL/100 g by oral gavage; Dex: adrx control group and they received IM dexamethasone 120 µg/kg/day and 0.1 mL/100 g of palm olein by oral gavage; Dex + ATT and Dex + PTT: annatto tocotrienol and palm tocotrienol groups, where they received IM dexamethasone 120 µg/kg/day and annatto tocotrienol and palm tocotrienol mixture 60 mg/kg/day by oral gavage, respectively, six days a week. Further, 120 µg/kg of dexamethasone (Sigma-Aldrich, Darmstadt, Denmark) was injected intramuscularly after being dissolved in palm olein (Sime Darby, Petaling Jaya, Malaysia). The dose and duration of treatments were based on a prior study, which established the dosage and course of treatment [25,26]. All the treatments were given daily for 2 months, and the rats were euthanized after completing 2 months of treatments. Given that mineralocorticoids have little effect on bone metabolism, no substitute was used [27].

A concentration of 60 milligrams per milliliter was obtained by dissolving 600 milligrams of annatto tocotrienol (American River Nutrition, Hadley, MA, USA) in 10 milliliters of palm olein. The rats were given an oral gavage of 0.1 mL/100 g body weight. All treatments were given for two months before the animals were euthanized.

The animals were housed in clean plastic conventional cages in an animal house that was dark at night and naturally light during the day. Each rat was placed in an individual cage. The rats were fed with rat pellets (Gold Coin, Klang, Malaysia), which were given ad libitum. While the sham-operated animals were given tap water, the adrenalectomized animals were given regular saline to drink ad libitum to make up for the salt loss caused by a mineralocorticoid deficiency. The animals were treated for two months before being anesthetized and euthanized.

Serum osteocalcin bone biomechanical strength, serum CTX, oxidative stress enzymes, gene expressions, and structural and static bone histomorphometry parameters were all examined.

### 4.2. Sample Collection

Blood samples were taken both before and after the treatment began. The blood samples were centrifuged for 15 min at 4 °C and 3000 rpm. The serum was kept in aliquots at −80 °C until analysis. Both femoral bones had their soft tissues removed. The right femoral bones were wrapped in gauze soaked in phosphate-buffered saline (PBS) and frozen at −80 °C prior to being used for biomechanical testing. The right femoral bones were sliced at the midshaft with a rotary blade (Black & Decker, Towson, Maryland, USA) to separate the proximal and distal sections. The distal part was cut longitudinally to separate the bones into medial and lateral pieces. The left femoral bones were examined for oxidative stress enzymes.

### 4.3. Measurement of Serum Bone Biochemical Markers

A total of eight samples per group were analyzed in these assays. Plasma samples were stored at −80 °C until analysis. Prior to the assay, samples were thawed at room temperature and mixed thoroughly. Serum levels of cross-linked C-telopeptide of type I collagen (CTX-I), a marker of bone resorption, were determined using a commercial ELISA kit (Uscn Lifescience Inc., Wuhan, China, Catalog E90665Ra), according to the manufacturer’s instructions. For the competitive inhibition ELISA technique, briefly, standards, controls, and samples were pipetted into pre-coated microplate wells, followed by incubation at 37 °C. After removal of unbound material, sequential incubations with Detection Reagent A and Detection Reagent B were performed, each followed by washing with the supplied buffer to remove excess reagents. Substrate solution was then added to allow color development in proportion to the amount of CTX-I bound. The reaction was terminated by the addition of the stop solution, and absorbance was measured at 450 nm using a microplate reader. A standard curve was generated from serial dilutions of the provided standards, and CTX-I concentrations in the samples were extrapolated using curve-fitting software (SoftMax Pro, Molecular Devices, San Jose, CA, USA). All samples were assayed in duplicate, and results were expressed as picograms per milliliter (pg/mL).

Serum osteocalcin was quantified using a commercial ELISA kit (Immunodiagnostic Systems Ltd., Tyne and Wear, UK, catalog AC-12F1), according to the manufacturer’s instructions. Briefly, samples, standards, and quality controls were assayed on pre-coated microplate wells, followed by sequential incubation with the provided detection reagents and washing steps to remove unbound material. Color development was produced by the kit substrate and terminated with stop solution; absorbance was measured using a microplate reader at the wavelength specified by the manufacturer. A standard curve was constructed from the supplied calibrators and sample concentrations interpolated using four-parameter logistic curve fitting. All samples were analyzed in duplicate and results reported in ng/mL. Assay performance was monitored using manufacturer-supplied and in-house quality controls.

### 4.4. Bone Biomechanical Test

The biomechanical properties of the femoral bones were assessed using the Instron Universal Testing Machine (model 5560, Instron, Canton, MA, USA) and the Bluehill 2 (Instron, Canton, MA, USA) software. Each femoral bone was placed in a three-point bending configuration on two lower supports that were 5 mm apart. This procedure was explained in a previous study [79]. The anterior surface was subjected to a force at mid-diaphysis, compressing it and putting tension on the posterior surface until the bone shattered. Plots of stress against strain and load against displacement were created using software that recorded the properties for load, displacement stress, and strain. The slope value of the load–displacement curve represented the femur’s modulus of elasticity. The bone mechanical test parameters can be divided into two main categories: intrinsic and extrinsic. While the extrinsic parameters—load, energy, and extension—measure the properties of the entire bone, the intrinsic parameters—stress, strain, and modulus of elasticity—measure the composition of the bone as explained by Turner 2002 [80].

### 4.5. Bone Histomorphometric Analysis

Eight samples from each experimental group were used for this test. Undecalcified bone samples were embedded in a mixture of Osteo Bed Resin Solution A (Polysciences Inc., Warrington, PA, USA) and Benzoyl Peroxide Plasticized (Catalyst) in the ratio of 100 mL of Osteo Bed Resin Solution A: 1.4 g of Benzoyl Peroxide Plasticized (Catalyst) for structural measurements, which included BV/TV, Tb.N, Tb.Th, and Tb.Sp. Using a microtome (Leica RM2155, Nussloch, Germany), the samples were sectioned at 7 μm thicknesses and stained using the Von Kossa technique. Briefly, acetone (HmBG Chemicals, Hamburg, Germany), decreasing ethanol concentrations (100%, 96%, and 70%), and water were used to treat bone sections. After being exposed to light for 20 min, sections were treated with 1% silver nitrate (Sigma, St. Louis, MO, USA), 2.5% sodium thiosulphate (Sigma, St. Louis, MO, USA), and increasing ethanol concentrations (70%, 96%, and 100%). For viewing under a light microscope (Olympus, Tokyo, Japan) with Image-PRO Plus software version 5.0.2.9, slides were cleaned using diethyl ether (HmBG Chemicals, Hamburg, Germany), dropped with dibutylphthalate polystyrene xylene (Sigma-Aldrich, Darmstadt, Denmark), and covered with a cover slip. Structural parameters of trabecular bone were assessed using an automated computer-assisted image analysis system (Video-Test Master software, St. Petersburg, Russia). The system was connected to a digital camera mounted on a light microscope to capture images of demineralized bone sections. Sections were stained with the von Kossa method, which renders trabecular bone dark brown against a white background, thereby facilitating image recognition and analysis. All measurements were performed at 50× magnification. For each bone sample, three histological slides were prepared, and pictures from three regions were randomly selected per slide for analysis. The mean value of these nine readings was recorded as the final measurement for each sample. The primary parameters obtained were trabecular bone area, total tissue area, and bone perimeter. These were expressed as Bone Volume (BV), Tissue Volume (TV), and Bone Surface (BS), respectively. From these primary measurements, secondary structural parameters were calculated according to the standardized formulas recommended by the ASBMR Histomorphometry Nomenclature Committee. These included the following:Bone Volume/Tissue Volume (BV/TV, %): BV ÷ TVTrabecular Thickness (Tb.Th, µm): BV ÷ (½BS)Trabecular Number (Tb.N,/mm): (BV/TV) ÷ Tb.ThTrabecular Separation (Tb.Sp, µm): (1 − Tb.Th) ÷ Tb.N

Histomorphometric parameter measurements were randomly performed at the metaphyseal region, which is located 3–7 mm from the lowest point of the growth plate and 1 mm from the lateral cortex, excluding the endocortical region. The selected area is the secondary spongiosa area, which is rich in trabecular bone. All parameters were measured according to the guidelines set by the American Society of Bone Mineral Research Histomorphometry Nomenclature Committee (1987) [81].

The bones were fixed in phosphate-buffered formalin and then decalcified for ten weeks using EDTA in order to assess static bone histomorphometry. Bones were cleaned with water and treated with increasing concentrations of ethanol (HmBG Chemicals, Hamburg, Germany), absolute toluene, and combined ethanol and toluene (1:1) (R&M Chemicals, Semenyih, Malaysia). Bones were then embedded, then sectioned into 5 μm thickness with a microtome (Leica, Wetzlar, Germany) after being embedded in paraffin wax (Leica Biosystems, Nussloch, Germany) three times for three hours each. the bone slices were stained using the Hematoxylin and Eosin (H&E) staining technique. Image-PRO Plus software version 5.0.2.9 was used to capture images of decalcified sections at 200× magnification using a light microscope (Olympus, Tokyo, Japan).

Counting of bone cellular histomorphometric parameters was performed by a blinded examiner using the Weibel Grid technique at the metaphysis’s secondary spongiosa region, which is situated 3–7 mm below the epiphyseal plate and 1 mm from the lateral cortex. Osteoblast number/bone area (Ob.N/BA, unit: number per bone area in mm), osteoclast number/bone area (Oc.N/BA, unit: number per bone area in mm^2^), eroded surface/bone surface (ES/BS, unit: %), osteoid surface/bone surface (OS/BS, unit: %), and osteoid volume/bone volume (OV/BV, unit: %) were the parameters derived from H&E staining. For each bone sample, three histological slides were prepared, and pictures from three regions were randomly selected per slide for analysis. The mean value of these nine readings was recorded as the final measurement for each sample. The procedure of bone histomorphometry had been described by Parfitt et al. [81].

Double fluorescence labeling was the method used to measure dynamic parameters. Unstained undecalcified bone samples were embedded in a mixture of Osteo Bed Resin Solution A (Polysciences Inc., Warrington, Pennsylvania, USA) and Benzoyl Peroxide Plasticized (Catalyst) in the ratio of 100 mL of Osteo Bed Resin Solution A: 1.4 g of Benzoyl Peroxide Plasticized (Catalyst) which has been sectioned was used for measuring these parameters. This was carried out to evaluate the mineralized surface (MS/BS), mineral appositional rate (MAR), and bone formation rate (BFR) of undecalcified femurs at 7-day intervals. An image analyzer (Eclipse 80i, Nikon) with Pro Plus 5.0 software (Media Cybernatics, Silver Spring, MD, USA) was linked to a fluorescent camera (Nikon, Tokyo, Japan) to capture micrographs. The single-labeled surface (sLS/BS) and double-labeled surface (dLS/BS), which were also a source of the parameters, and calculation were performed by the Weiber et al. technique.

Dynamic parameters were assessed on undecalcified, unstained bone sections using the point-counting technique. Fluorescent images were captured with an image analyzer (Eclipse 80i, Nikon) with Pro Plus 5.0 software (Media Cybernatics, Silver Spring, MD, USA) was linked to a fluorescent camera (Nikon, Tokyo, Japan) to capture micrographs at 200× magnification and displayed on a computer screen. To obtain the fluorescent labeling on the trabecular surfaces, the rats were given injections of calcein (20 mg/kg body weight) at 9 and 2 days before their euthanasia. This was carried out to evaluate the MS/BS, MAR, and BFR of undecalcified femurs at 7-day intervals. For each bone sample, three histological slides were prepared, and pictures from three regions were randomly selected per slide for analysis. The mean value of these nine readings was recorded as the final measurement for each sample. A Weibel grid was applied, and the following measurements were obtained:Single-Labeled Surface per Bone Surface (sLS/BS, %): proportion of bone surface with a single fluorescent label.Double-Labeled Surface per Bone Surface (dLS/BS, %): proportion of bone surface with two fluorescent labels.Mineralized Surface per Bone Surface (MS/BS, %): calculated as:$$\mathrm{MS}/\mathrm{BS}=\frac{\mathrm{dLS}+1/2\mathrm{sL}}{\mathrm{BS}}$$

A semi-automated method combining Image Pro-Express software (Media Cybernetics, Rockville, MA, USA) with manual verification was used to calculate inter-label distances. The MAR (µm/day) was derived by dividing the average inter-label distance by seven days. The BFR/BS (µm^3^/µm^2^/day) was then calculated as follows:$$\mathrm{BFR}=\frac{\mathrm{dLS}+1/2\mathrm{sLS}}{\mathrm{BS}}\;\times \mathrm{MAR}$$

### 4.6. Gene Expression Measurements

Trabecular bone tissue was harvested from the distal region of the left femurs. For this analysis, eight samples were obtained from each group. Approximately 5 mg of tissue (equivalent to the size of a rice grain) was collected per sample and placed into individual vials containing magnetic beads. Tissue homogenization was performed using the QG Sample Preparation Kit for Tissue (Affymetrix Thermo Fisher Scientific, Santa Clara, CA, USA, catalog number QS0104). Briefly, 3 µL of homogenizing solution and 3 µL of proteinase K were added to each vial. Samples were subjected to two cycles of mechanical disruption (15–30 s each) using a cell rupture apparatus, followed by a brief spin. The homogenates were then incubated at 65 °C for 30–60 min in a water bath, centrifuged at 13,000 rpm for 10 min at room temperature, and the resulting supernatants were transferred into microcentrifuge tubes and stored at −80 °C until analysis.

Gene expression was quantified using the QuantiGene Plex 2.0 Assay (Affymetrix Thermo Fisher Scientific, Santa Clara, CA, USA, catalog number QS0104), following the manufacturer’s instructions. This assay combines branched DNA (bDNA) signal amplification technology with Luminex xMAP^®^ bead-based multiplexing, enabling the simultaneous detection of multiple RNA targets without reverse transcription or PCR. Target-specific oligonucleotide probe sets—comprising capture extenders, label extenders, and blockers—were designed by the manufacturer. These hybridize directly to target RNA molecules, which are then immobilized onto fluorescent capture beads. Signal amplification is achieved through hybridization of branched DNA amplifier molecules bearing multiple biotin sites, which are subsequently labeled with streptavidin-conjugated R-phycoerythrin (SAPE).

Fluorescent signals were acquired using a Luminex instrument (Bio-Rad, Hercules, CA, USA, catalog number 171000201). For each target gene, the median fluorescence intensity (MFI) was recorded, reflecting the abundance of captured RNA. Expression levels of target genes were normalized against the housekeeping gene glyceraldehyde-3-phosphate dehydrogenase (GAPDH) to control for input variation.

The QuantiGene Plex 2.0 assay is a branched-DNA (bDNA) signal-amplification method that quantifies specific RNA targets without reverse transcription or PCR. For each target transcript, a proprietary probe set composed of capture extenders (CEs), label extenders (LEs), and blockers (BLs) hybridizes directly to the target RNA. CE oligonucleotides mediate capture of the target–probe complex to bead-bound capture sequences, while LEs provide binding sites for branched DNA amplifier structures bearing biotinylated probe binding sites. Streptavidin-phycoerythrin (SAPE) binds to the biotinylated amplifier structures to generate a fluorescent signal.

Briefly, probe sets and samples were combined with the multiplexed capture-bead array in assay plates and hybridized to allow target–probe binding. Following hybridization, a sequence of signal amplification steps was performed using the kit-supplied amplifier and label reagents, producing branched DNA structures that link the captured target to biotin moieties. After final washing to remove unbound reagents, SAPE was added to label bound amplifiers. Fluorescent signals were acquired on a Luminex flow cytometer, which discriminates bead identity and measures reporter fluorescence associated with each bead population. For each bead region (target), the instrument reports a median fluorescence intensity (MFI), which is proportional to the abundance of the corresponding RNA target in the sample.

All samples and controls were assayed in technical duplicates. Each plate included a full set of kit calibrators, negative/background controls, and positive controls supplied by the manufacturer; in addition, internal control probes (housekeeping genes) and exogenous spike-in controls were included to monitor assay performance and to enable normalization. Plates were run with standardized instrument settings, and bead counts per analyte/well were monitored to ensure sufficient events for reliable statistics.

### 4.7. Measurement of Lipid Peroxidation and Oxidative Stress Enzymes

A total of eight samples per group were analyzed in these assays. Left distal femoral trabecular bone specimens were freed of adhering soft tissue and rinsed thoroughly in ice-cold phosphate-buffered saline (PBS; pH 7.4) to remove blood and surface contaminants. Bone tissue was then transferred to ice-cold extraction buffer (50 mM Tris–HCl, pH 7.5; 5 mM EDTA; 1 mM DTT). Buffer volume was adjusted to 5–10 mL per gram of wet tissue. All subsequent manipulations were performed on ice or at 4 °C to minimize enzymatic degradation.

Tissues were mechanically disrupted using a chilled tissue grinder until a uniform homogenate was obtained. Homogenates were centrifuged at 10,000× *g* for 15 min at 4 °C, and the resulting supernatants were collected, aliquoted to avoid repeated freeze–thaw cycles, and stored at −80 °C until assay. Insoluble pellets were discarded. Prior to biochemical assays, the protein concentration in each homogenate was determined (e.g., bicinchoninic acid or Bradford assay) and used to normalize enzymatic activities and lipid peroxidation results (expressed per mg protein). Where appropriate, protease and phosphatase inhibitor cocktails may have been included in the extraction buffer to preserve labile analytes; if used, reagent compositions and final concentrations are reported. All enzymatic assays (SOD, CAT, GPX) and MDA measurements were performed according to the manufacturers’ instructions, with samples assayed in technical duplicate. Assay results were quality-checked for intra- and inter-assay variability, and values falling outside of the predefined acceptance criteria were re-assayed or excluded. Data are reported as mean ± SD (or median [IQR]), and statistical comparisons were performed on protein-normalized values. ELISA kits (Cayman Chemical Company, Ann Arbor, MI, USA) were utilized to assess SOD (catalog number 706002)**,** a colorimetric activity assay, CAT (catalog number 707002), a colorimetric assay and GPX catalog number 703102, a coupled enzymatic activity assay. Malondialdehyde (MDA) was used to measure the lipid peroxidation activity. The Lipid Peroxidation (MDA) Colorimetric/Fluorometric Assay Kit, catalog K739-100 kit from BioVision Incorporated (Milpitas, CA, USA), was used to measure the MDA level.

### 4.8. Statistical Analysis

The statistical software used for data analysis was the Statistical Package for Social Sciences (SPSS) version 20.1.2 (IBM, Armonk, NY, USA). The data were tested for normality using the Kolmogorov–Smirnov test. Since the groups were found to be normally distributed, the data were analyzed using parametric statistics, i.e., the ANOVA test followed by the Tukey post hoc test for comparison between treatment groups. *p* values < 0.05 were taken as significant. Data were presented as mean + standard error of the mean (SEM).

## 5. Conclusions

The study’s findings supported annatto tocotrienol’s antioxidant properties and its capacity to shield bone from free radical damage. This could create a new avenue for preventing osteoporosis brought on by glucocorticoids and the fractures that ensue from it. Given its favorable effects and safety record, tocotrienol produced from annatto is a promising natural supplement for preventing osteoporosis brought on by glucocorticoids. However, more research is required to fully understand the mechanisms at play.

## Figures and Tables

**Figure 1 ijms-26-10206-f001:**
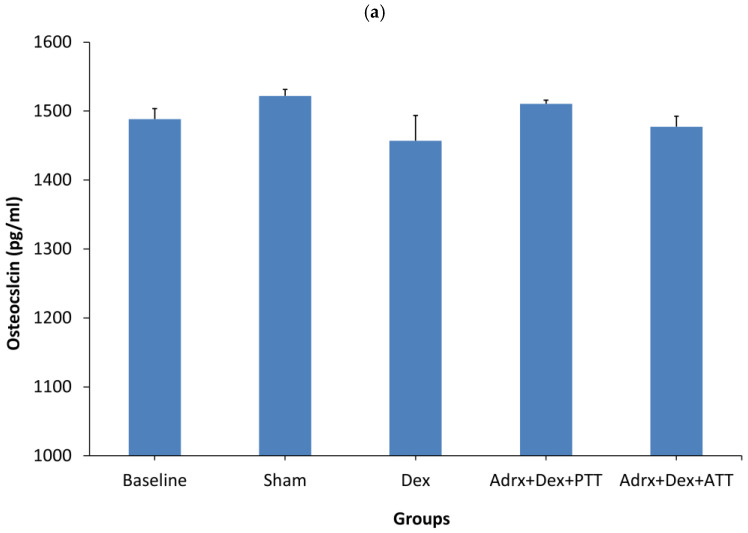
Expression of protein related to bone formation. (**a**) Osteocalcin; (**b**) Serum CTX (C-terminal telopeptide of type I collagen). Baseline: Rats were euthanized without any treatment, Sham: sham-operated group, given vehicle palm olein 0.05 mL/100 g by intramuscular (IM) injection and 0.1 mL/100 g by oral gavage, Dex: adrenalectomized (adrx) control group and given IM dexamethasone 120 µg/kg/day and 0.1 mL/100 g of palm olein by oral gavage; Dex + ATT and Dex + PTT: annatto tocotrienol and palm tocotrienol groups where they received IM dexamethasone 120 µg/kg/day and annatto tocotrienol and palm tocotrienol mixture 60 mg/kg/day by oral gavage, respectively, six days a week. N = 8 for each group. Data presented as mean + SEM. * indicates significant difference between groups at *p* < 0.05. Osteocalcin levels showed no significant differences among groups compared to baseline. Post-treatment CTX levels increased significantly in all groups relative to pre-treatment values. The Dex+PTT and Dex+ATT groups demonstrated significantly lower post-treatment CTX levels than the Dex group.

**Figure 2 ijms-26-10206-f002:**
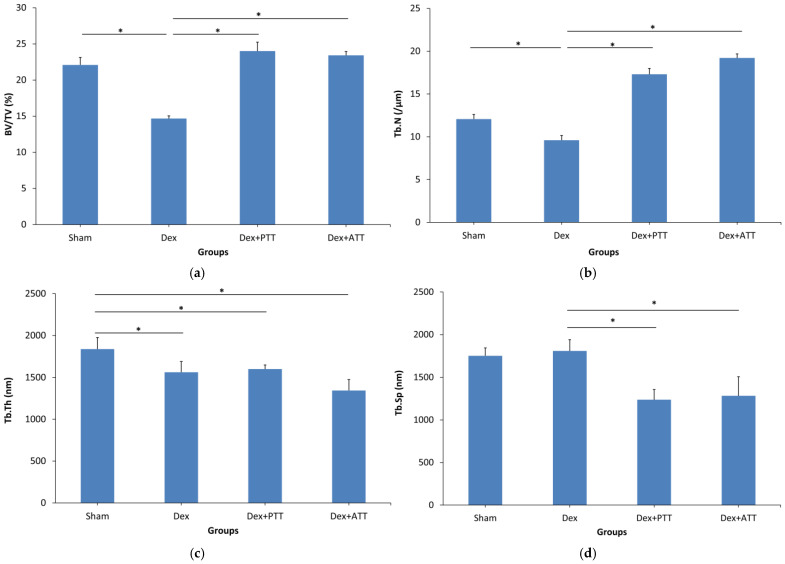
Structural histomorphometric parameters after 8 weeks of treatment. (**a**) Bone volume (BV/TV); (**b**) Trabecular number (Tb.N); (**c**) Trabecular thickness (Tb.Th); (**d**) Trabecular separation (Tb.Sp). Sham: sham-operated group, given vehicle palm olein 0.05 mL/100 g by intramuscular (IM) injection and 0.1 mL/100 g by oral gavage, Dex: adrenalectomized (adrx) control group and given IM dexamethasone 120 µg/kg/day and 0.1 mL/100 g of palm olein by oral gavage; Dex + ATT and Dex + PTT: annatto tocotrienol and palm tocotrienol groups that received IM dexamethasone 120 µg/kg/day and annatto tocotrienol and palm tocotrienol mixture 60 mg/kg/day by oral gavage, respectively, six days a week. N = 8 for each group. Data presented as mean ± SEM. * indicates significant difference between groups at *p* < 0.05. The Dex group showed significantly lower BV/TV, Tb.N, and Tb.Th, with no significant change in Tb.Sp compared to the Sham group. Both Dex+PTT and Dex+ATT groups demonstrated significant increases in BV/TV and Tb.N, and a significant reduction in Tb.Sp, with no significant change in Tb.Th.

**Figure 3 ijms-26-10206-f003:**
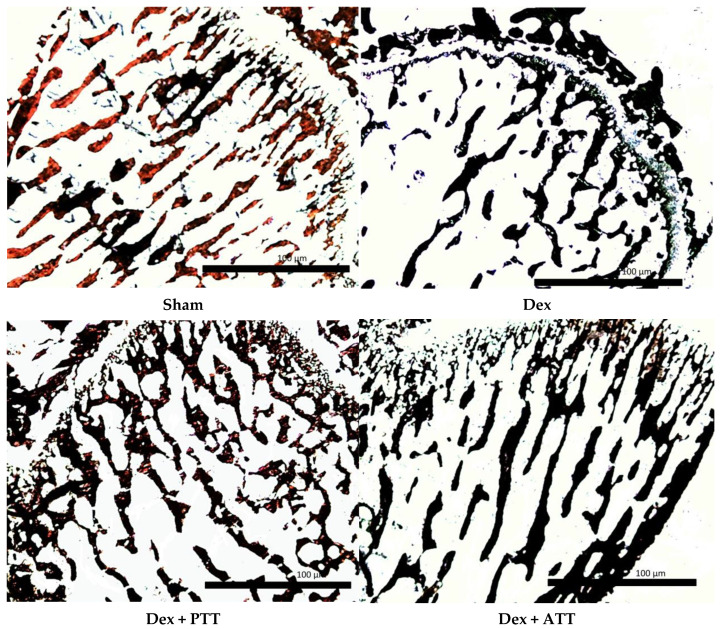
Photomicrograph of undecalcified trabecular bones for structural histomorphometry parameters using von Kossa staining (200× magnification). Sham: sham-operated group, given vehicle palm olein 0.05 mL/100 g by intramuscular (IM) injection and 0.1 mL/100 g by oral gavage, Dex: adrenalectomized (adrx) control group and given IM dexamethasone 120 µg/kg/day and 0.1 mL/100 g of palm olein by oral gavage; Dex + ATT and Dex + PTT: annatto tocotrienol and palm tocotrienol groups where they received IM dexamethasone 120 µg/kg/day and annatto tocotrienol and palm tocotrienol mixture 60 mg/kg/day by oral gavage, respectively, six days a week.

**Figure 4 ijms-26-10206-f004:**
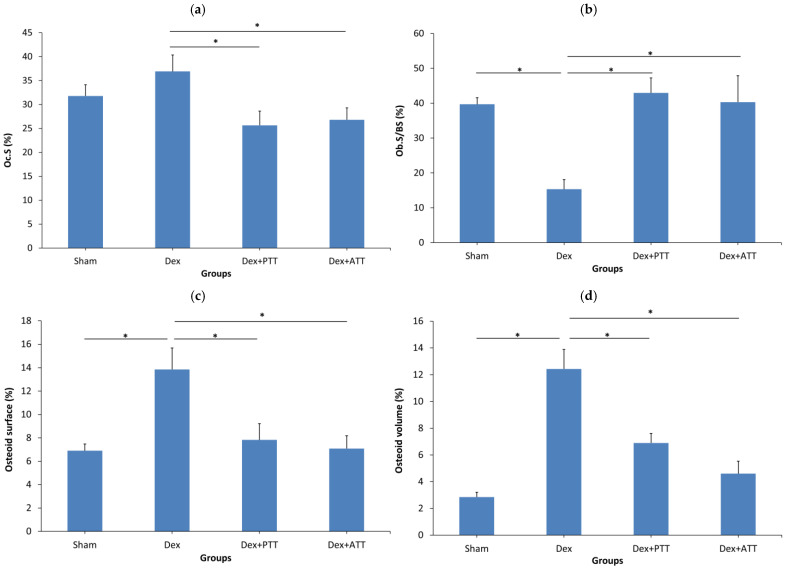
Static trabecular histomorphometry parameters after 8 weeks of treatment. (**a**) Osteoclast surface (Oc.S/BS); (**b**) Osteoblast surface (Ob.S/BS); (**c**) Osteoid surface (OS/BS); (**d**) Osteoid volume (OV/BV). Sham: sham-operated group, given vehicle palm olein 0.05 mL/100 g by intramuscular (IM) injection and 0.1 mL/100 g by oral gavage, Dex: adrenalectomized (adrx) control group and given IM dexamethasone 120 µg/kg/day and 0.1 mL/100 g of palm olein by oral gavage; Dex + ATT and Dex + PTT: annatto tocotrienol and palm tocotrienol groups where they received IM dexamethasone 120 µg/kg/day and annatto tocotrienol and palm tocotrienol mixture 60 mg/kg/day by oral gavage, respectively, six days a week. N = 8 for each group. Data presented as mean + SEM. * indicates significant difference between groups at *p* < 0.05. The Dex group showed significantly lower Ob.S/BS, and significant increase in osteoid surface and osteoid volume with no significant change in Oc.S/BS compared to the Sham group. Both Dex+PTT and Dex+ATT groups demonstrated significant increases in Ob.S/BS and a significant reduction in Oc.S/BS.

**Figure 5 ijms-26-10206-f005:**
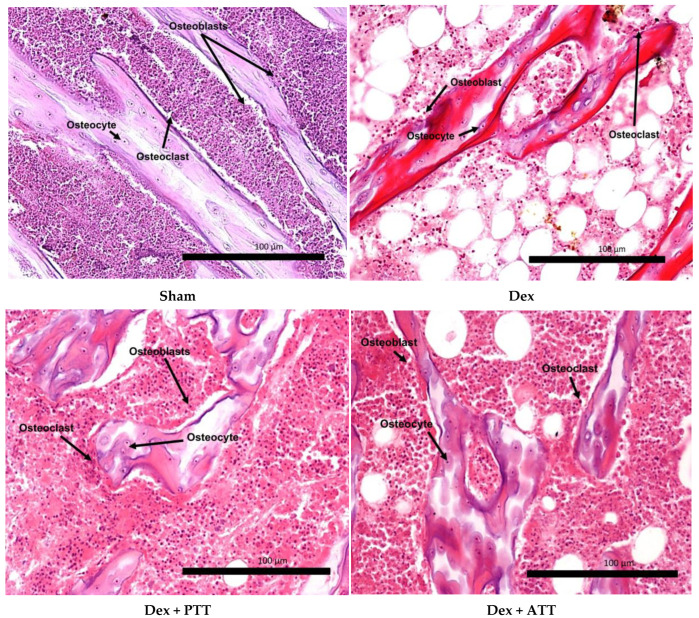
Hematoxylin and eosin stain of the decalcified bone of the distal femur at 200× magnification, showing trabecular bone with osteoblast and osteoclast. Sham: sham-operated group, given vehicle palm olein 0.05 mL/100 g by intramuscular (IM) injection and 0.1 mL/100 g by oral gavage, Dex: adrenalectomized (adrx) control group and given IM dexamethasone 120 µg/kg/day and 0.1 mL/100 g of palm olein by oral gavage; Dex + ATT and Dex + PTT: annatto tocotrienol and palm tocotrienol groups where they received IM dexamethasone 120 µg/kg/day and annatto tocotrienol and palm tocotrienol mixture 60 mg/kg/day by oral gavage, respectively, six days a week. Dexamethasone-treated rats (Dex) showed a marked reduction in osteoblast surface (Ob.S/BS) and an increase in osteoclast surface (Oc.S/BS) compared to the Sham group. Conversely, rats supplemented with tocotrienol (Dex + ATT and Dex + PTT) demonstrated higher Ob.S/BS and lower Oc.S/BS.

**Figure 6 ijms-26-10206-f006:**
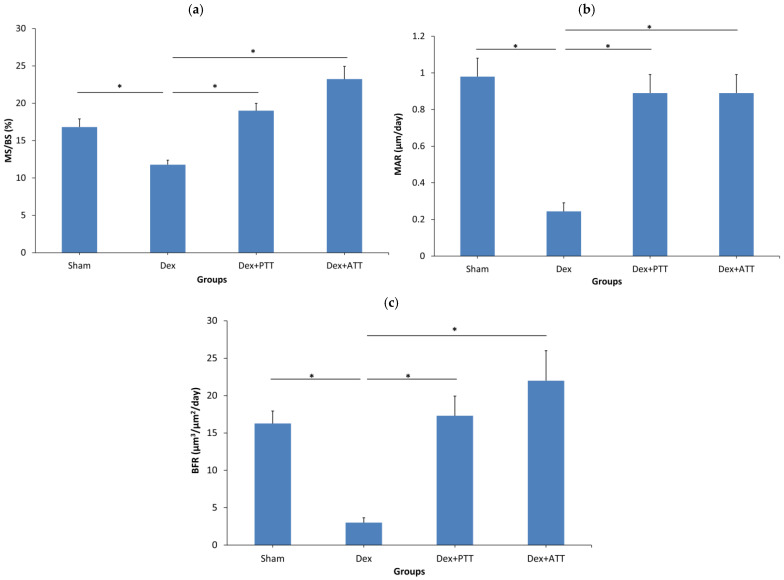
Dynamic trabecular histomorphometry parameters after 8 weeks of treatment. (**a**) Mineralized surface (MS/BS). (**b**) Mineral apposition rate (MAR). (**c**) Bone formation rate (BFR). Sham: sham-operated group, given vehicle palm olein 0.05 mL/100 g by intramuscular (IM) injection and 0.1 mL/100 g by oral gavage, Dex: adrenalectomized (adrx) control group and given IM dexamethasone 120 µg/kg/day and 0.1 mL/100 g of palm olein by oral gavage; Dex + ATT and Dex + PTT: annatto tocotrienol and palm tocotrienol groups where they received IM dexamethasone 120 µg/kg/day and annatto tocotrienol and palm tocotrienol mixture 60 mg/kg/day by oral gavage, respectively, six days a week. N = 8 for each group. Data presented as mean + SEM. * indicates significant difference between groups at *p* < 0.05. The Dex group showed significantly lower MS/BS, MAR and BFR compared to the Sham group. Both Dex+PTT and Dex+ATT groups demonstrated significant increases in MS/BS, MAR and BFR compared to the Dex group.

**Figure 7 ijms-26-10206-f007:**
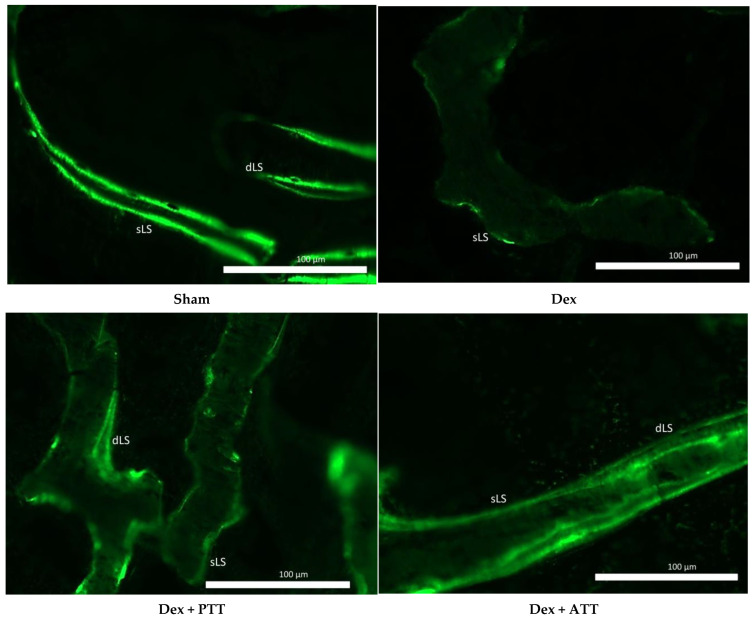
Photomicrograph of undecalcified sections of calcein-labeled trabecular bone of the distal femur for dynamic histomorphometry parameters (200× magnification). Sham: sham-operated group, given vehicle palm olein 0.05 mL/100 g by intramuscular (IM) injection and 0.1 mL/100 g by oral gavage, Dex: adrenalectomized (adrx) control group and given IM dexamethasone 120 µg/kg/day and 0.1 mL/100 g of palm olein by oral gavage; Dex + ATT and Dex + PTT: annatto tocotrienol and palm tocotrienol groups where they received IM dexamethasone 120 µg/kg/day and annatto tocotrienol and palm tocotrienol mixture 60 mg/kg/day by oral gavage, respectively, six days a week. Rats treated with dexamethasone (Dex) exhibited a reduction in both single-labeled surface (sLS) and double-labeled surface (dLS) compared to the Sham group. In contrast, rats supplemented with tocotrienol (Dex + ATT and Dex + PTT) demonstrated higher sLS and dLS.

**Figure 8 ijms-26-10206-f008:**
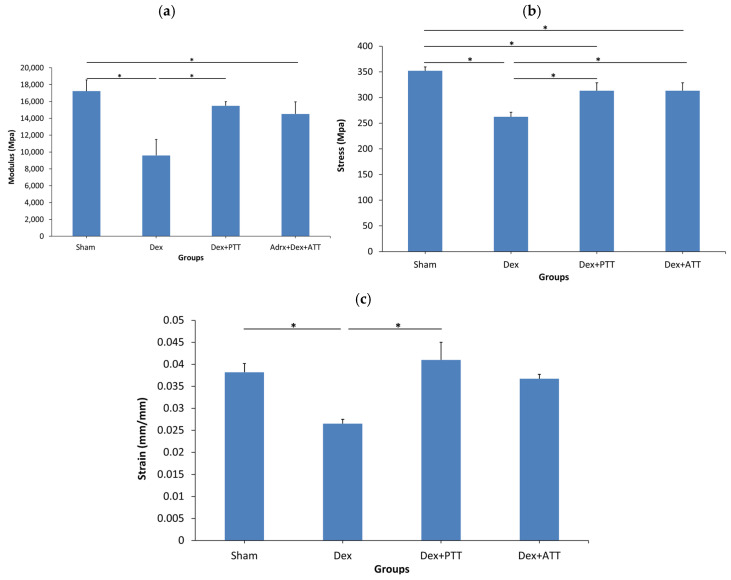
Intrinsic parameters of bone biomechanical strength. (**a**) Young’s Modulus. (**b**) Stress at maximum flexure load. (**c**) Strain at maximum flexure load. SHAM: sham-operated group, given vehicle palm olein 0.05 mL/100 g by intramuscular (IM) injection and 0.1 mL/100 g by oral gavage. Dex: adrenalectomized (adrx) control group and they received IM dexamethasone 120 µg/kg/day and 0.1 mL/100 g of palm olein by oral gavage; Dex + ATT and Dex + PTT: annatto tocotrienol and palm tocotrienol groups where they received IM dexamethasone 120 µg/kg/day and annatto tocotrienol and palm tocotrienol mixture 60 mg/kg/day by oral gavage, respectively, six days a week. N = 8 for each group. Data presented as mean + SEM. * indicates significant difference between groups at *p* < 0.05. The Dex group showed significantly lower modulus, stress and strain compared to the Sham group. Dex+PTT group demonstrated significant increases in modulus, stress and strain compared to the Dex group. Dex + ATT showed similar effects except for the modulus.

**Figure 9 ijms-26-10206-f009:**
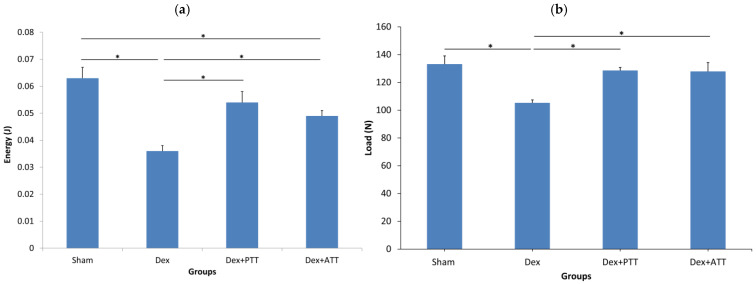
Extrinsic parameters of bone biomechanical strength. (**a**) Energy at maximum flexure load. (**b**) Maximum flexure load. (**c**) Maximum flexure extension. SHAM: sham-operated group, given vehicle palm olein 0.05 mL/100 g by intramuscular (IM) injection and 0.1 mL/100 g by oral gavage. Dex: adrenalectomized (adrx) control group and they received IM dexamethasone 120 µg/kg/day and 0.1 mL/100 g of palm olein by oral gavage; Dex + ATT and Dex + PTT: annatto tocotrienol and palm tocotrienol groups where they received IM dexamethasone 120 µg/kg/day and annatto tocotrienol and palm tocotrienol mixture 60 mg/kg/day by oral gavage, respectively, six days a week. N = 8 for each group. Data presented as mean + SEM. * indicates significant difference between groups at *p* < 0.05. The Dex group showed significantly lower energy, load and maximum extension compared to the Sham group. Both Dex+PTT and Dex+ATT groups demonstrated significant increases in energy and load with no significant changes to the maximum extension compared to the Dex group.

**Figure 10 ijms-26-10206-f010:**
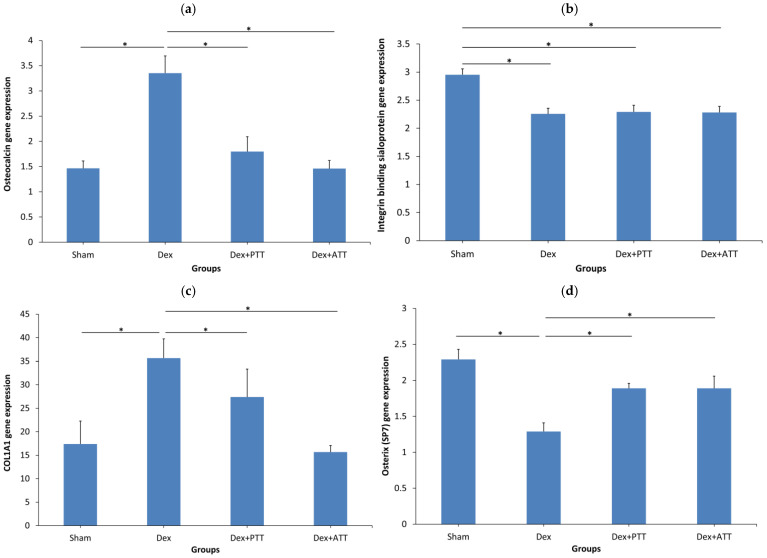
Expression of genes related to bone formation. SHAM: sham-operated group, given vehicle palm olein 0.05 mL/100 g by intramuscular (IM) injection and 0.1 mL/100 g by oral gavage. Dex: adrenalectomized (adrx) control group and they received IM dexamethasone 120 µg/kg/day and 0.1 mL/100 g of palm olein by oral gavage; Dex + ATT and Dex + PTT: annatto tocotrienol and palm tocotrienol groups where they received IM dexamethasone 120 µg/kg/day and annatto tocotrienol and palm tocotrienol mixture 60 mg/kg/day by oral gavage, respectively, six days a week. Data presented as mean + SEM. N = 8 for each group. * indicates significant difference between groups at *p* < 0.05. The Dex group showed significantly higher expression of osteocalcin (**a**) and COL1A1 (**c**) and lower IBSP (**b**) and osterix (**d**) gene expression compared to the Sham group. Both Dex+PTT and Dex+ATT groups demonstrated significant lower expression of osteocalcin (**a**) and COL1A1 (**c**) and higher osterix (**d**) gene expression, while the IBSP expression remained unchanged compared to the Dex group.

**Figure 11 ijms-26-10206-f011:**
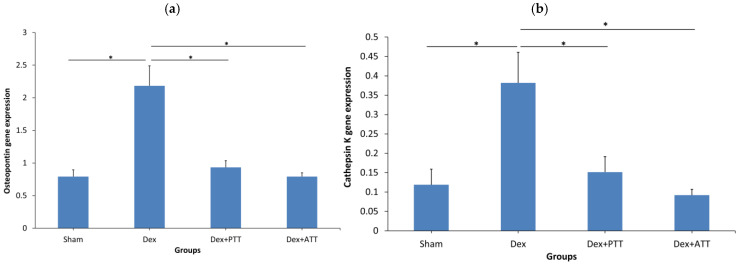
Expression of genes related to bone resorption. SHAM: sham-operated group, given vehicle palm olein 0.05 mL/100 g by intramuscular (IM) injection and 0.1 mL/100 g by oral gavage. Dex: adrenalectomized (adrx) control group and they received IM dexamethasone 120 µg/kg/day and 0.1 mL/100 g of palm olein by oral gavage; Dex + ATT and Dex + PTT: annatto tocotrienol and palm tocotrienol groups where they received IM dexamethasone 120 µg/kg/day and annatto tocotrienol and palm tocotrienol mixture 60 mg/kg/day by oral gavage, respectively, six days a week. N = 8 for each group. Data presented as mean + SEM. * indicates significant difference between groups at *p* < 0.05. The Dex group showed significantly higher expression of osteopontin (**a**), cathepsin K (**b**) and RANKL/OPG (**c**) gene expression compared to the Sham group. Both Dex+PTT and Dex+ATT groups demonstrated significant lower expression of osteopontin (**a**) and cathepsin K (**b**), while the RANKL/OPG ratio remained unchanged.

**Figure 12 ijms-26-10206-f012:**
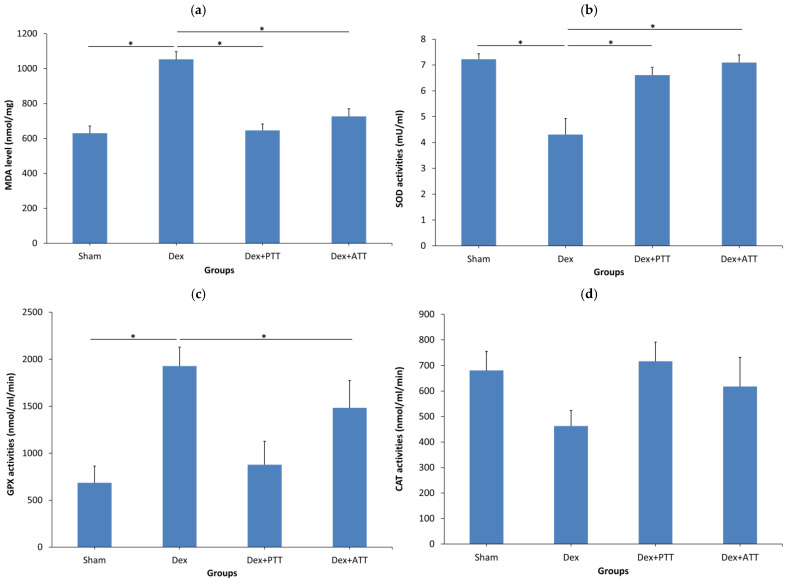
Lipid peroxidation, superoxide dismutase (SOD), glutathione peroxidase (GPX), and catalase (CAT) activities in the bone. Data presented as mean + SEM. Same letters indicate significant difference between treatment groups at *p* < 0.05, SHAM: sham-operated group, given vehicle palm olein 0.05 mL/100 g by intramuscular (IM) injection and 0.1 mL/100 g by oral gavage, Dex: adrenalectomized (adrx) control group and they received IM dexamethasone 120 µg/kg/day and 0.1 mL/100 g of palm olein by oral gavage; Dex + ATT and Dex + PTT: annatto tocotrienol and palm tocotrienol groups where they received IM dexamethasone 120 µg/kg/day and annatto tocotrienol and palm tocotrienol mixture 60 mg/kg/day by oral gavage, respectively, six days a week. N = 8 for each group. Data presented as mean + SEM. * indicates significant difference between groups at *p* < 0.05. The Dex group showed significant increase in the MDA (**a**) level and GPX (**c**) activity, and reduced SOD activity (**b**) compared to the Sham group. The CAT activity (**d**) showed no significant changes. Both Dex+PTT and Dex+ATT groups showed significant decrease to the MDA (**a**) level and increase to the SOD activity (**b**). Only Dex+PTT group showed significant decrease to the GPX group (**b**). CAT activity showed no significant difference.

**Figure 13 ijms-26-10206-f013:**
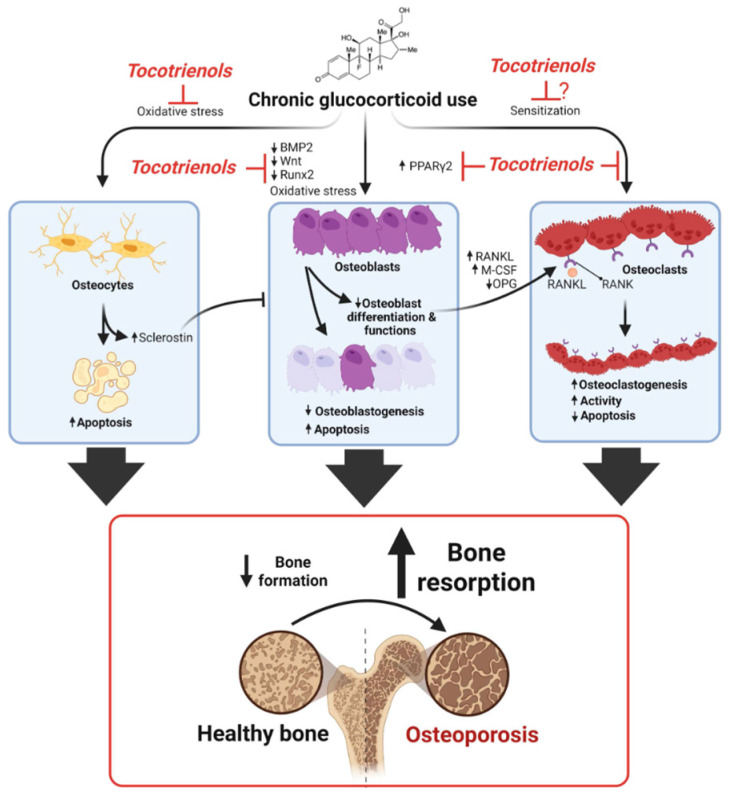
Overview of tocotrienols mediating osteoprotective effects on glucocorticoid-induced bone loss. Glucocorticoids are reported to reduce osteoblastogenesis by enhancing PPARγ2 and suppressing BMP2/Wnt/β-catenin/Runx2 signaling via oxidative stress and transrepression mechanisms. Glucocorticoid-mediated oxidative damage also triggers osteoblast and osteocyte apoptosis. In addition, glucocorticoids stimulate osteoblasts and bone stromal cells to produce more RANKL and M-CSF, which enhance osteoclast proliferation (osteoclastogenesis) and activity. These result in increased bone resorption and decreased bone formation. Tocotrienols inhibit glucocorticoid-mediated bone loss through multiple mechanisms. As potent antioxidants, they reduce glucocorticoid-induced oxidative stress, thereby protecting osteoblasts and osteocytes from apoptosis while preventing excessive osteoclast activation. Tocotrienols also enhance BMP2/Wnt/β-catenin/Runx2 and suppress PPARγ2 signaling pathways, which promote osteoblastogenesis and suppress adipogenesis. In addition, tocotrienols, known HMG-CoA reductase inhibitors, reduce protein prenylation of small GTPases, thus indirectly disrupting osteoclast function and bone resorption. Created in BioRender by Pang, K. (2025) [https://BioRender.com/nj7598h (assessed: 10/10/2025)]. Abbreviations: BMP2, bone morphogenetic protein 2; M-CSF, macrophage colony stimulating factor; OPG, osteoprotegerin; PPARγ2, peroxisome proliferator-activated receptor gamma 2; RANK, receptor activator of nuclear factor κB; RANKL, RANK ligand; Runx2, Runt-related transcription factor 2; Wnt, wingless/integrated signaling pathway.

## Data Availability

Data are available at reasonable request from the corresponding author.

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
