# Peer review of "Comparative Bone-Protective Effects of Tocotrienol Isomers from Palm and Annatto in Dexamethasone-Induced Osteoporotic Male Rats"

_ijms, 2025, doi:10.3390/ijms262010206_

Round 1
Reviewer 1 Report
Comments and Suggestions for Authors
Please, find the attached file.

Author Response
We are pleased to resubmit our revised manuscript entitled “Comparative Analysis of Pure Tocotrienol and Tocotrienol Mixture: Differential Effects of Palm and Annatto Tocotrienol on Bone Health in Glucocorticoid-Induced Osteoporosis” (Manuscript ID: ijms-3818621) for consideration in. We are grateful to you and the reviewers for your constructive comments and suggestions, which have helped us to improve the quality and clarity of our work.
In this revised version, we have carefully addressed all the reviewers’ comments as much as possible. A detailed point-by-point response to each comment is included as a separate document, highlighting the changes made. All modifications in the manuscript are marked [highlighted in yellow for your convenience.
We believe that these revisions have substantially strengthened our manuscript, and we hope that it will now be suitable for publication in IJMS.
Thank you for your time and consideration. We look forward to your response. We remain open to addressing any additional feedback that may arise
Sincerely,
Elvy Suhana Mohd Ramli
On behalf of all co-authors

Reviewer 2 Report
Comments and Suggestions for Authors
The paper titled ( Comparative Analysis of Pure Tocotrienol and Tocotrienol Mixture: Differential Effects of Palm and Annatto Tocotrienol on Bone Health in Glucocorticoid-Induced Osteoporosis ) by Authors Elvy Suhana Mohd Ramli and her colleagues is an experimental study using an animal model of glucocorticoid induced osteoporosis in male rats . The authors used biochemical and histomorphometric methods for evaluation of the effect of the tested agents and statistical analysis was done to compare Palm and Annatto Tocotrienol on Bone Health. However, the authors did not give a sharp conclusion which preparation was better for its bone preserving effect. The paper should be extensively revised
The methods used in carrying out the study are adequate. The results are documented and clarified . The discussion appears clear enough. Many studies are cited.
A main concern in this paper is the huge % of plagiarism 32% and 16% from one single source included in methods and even in Figure legends!! This must be solved before any further processing of this paper.
Actually the paper needs extensive revisions due to high plagiarism and check for the use of academic English.
Here, I am listing some recommendations for improving the topic. please find these comments & provide a point to point reply and highlight the changes in the file and indicate at what page & line we can follow every change.
1 - Title: Comparative Analysis of Pure Tocotrienol and Tocotrienol Mixture: Differential Effects of Palm and Annatto Tocotrienol on Bone Health in Glucocorticoid-Induced Osteoporosis ) contains redundant parts and needs revision
2- Title: mention that this is a rat study
3- Title: Glucocorticoid should be replaced by the specific name of drug/model used in the study
4 - Abstract must be amended by some numerical values for key findings from the study. I think also it is longer than required
5 - Key words: ( Annatto tocotrienol; biomechanical strength; bone histomorphometry; glucocorticoids; osteoporosis; oxidative stress.) can be better described
Authors should mention ( male rat ) and the same comment for title (dexamethasone) instead of (glucocorticoid)
6 - Introduction: too long and although introduces the items of the study well, did not explore the rational or novelty of the study.
It should be shortened tp about 700 words to be more concrete
7 - AIm of the study should be clear and clarify what was the aim and how authors acheived it
8 - Results
- Every abbreviation in figures should be explained in the figure legend to be self-explanatory & stands alone.
- - Mention "n" in each illustration individually
- Results in Figure 3 & 5 are of low value as authors did not refer to the pathologic features in osteoporosis rats anddi not use arrows or similar tools to refer to pathology
- How many sections were taken from each animal & how many images were taken from each section?
9 - Methods
- Experimental design : please give references and rationale for the selected doses of drugs and preparations
Please give a title for study design & describe the groups in details and regimens (how much, how frequent and how long) in a clear way
– Why you decided to do this study in male rats ?
mention the number of rats in each group? how many experimental groups in the study ? and how many total animals, please indicate in a clear way
- What was the age and weight of the animals at the begin of the study
- Authors should give the source of chemicals, kits and antibodies completely and consistently (code, company, town, state and country) & version for software
- Statistical tests and sample sizes for each experiment should be explicitly mentioned in the methods and figure legends.
- Authors have to check the normality of distribution of the results by a suitable post hoc test (such as Shapiro-Wilk test or K-S test) before deciding to choose certain ANOVA. If the normality test indicated normal dist of the data, so use one-way ANOVA, if not, use non parametric ANOVA. In all cases choose a suitable post-hoc test
- Authors should confirm in methods that "every possible comparison between the study groups was considered" and apply this in results
In methods, Mention in details the housing conditions and how authors were keen to reduce animal suffering
Animal details and housing should be separate from the experiment design.
Animal details and housing should be written in details (cage type, number per cage, food, dark light cycles, how minimized animal suffering...etc) .
Methods in general lacks references
10- General
- Use appropriate abbreviations for minutes, seconds...etc
- Ensure every abbreviation is explained at the first appearance in abstract & then in the body text
-Please write the limitations of this study and future directions after which
With the above revisions, I believe your manuscript will make a valuable contribution to the field. I encourage you to address these suggestions to improve the clarity and overall impact of your paper.
Author Response

(The authors gave the same response as above.)

Round 2
Reviewer 1 Report
Comments and Suggestions for Authors
The Authors addressed all comments and improved the manuscript considering the remarks. I believe that the manuscript may be published in its current form.
A couple of typos:
- Line 337: lost “N”
The Discussion section. Please check literature reports. For example:
- Lines 353-354 “(Baek 353 et al., 2010; Parhami et al., 1997).”
- Line 381 “(Niki et al., 2005)”
Author Response
Dear reviewer,
On behalf of the authors, I am resubmitting the second revision of the manuscript entitled ‘Comparative Bone-Protective Effects of Tocotrienol Isomers from Palm and Annatto in Dexamethasone-Induced Osteoporotic Male Rats’ Thank you for your valuable suggestions and comments. We tried our best to answer and respond to comments. We hope they meet your expectation and we are happy to respond to further suggestions. The changes are highlighted in yellow.
Best regards
Elvy Suhana Mohd Ramli
Corresponding author

Reviewer 2 Report
Comments and Suggestions for Authors
The revised version of paper titled (Comparative Analysis of Pure Tocotrienol and Tocotrienol Mixture: Differential Effects of Palm and Annatto Tocotrienol on Bone Health in Glucocorticoid-Induced Osteoporosis) by Authors Elvy Suhana Mohd Ramli et al.
1- Introduction is too too long, must be shortened to be more concrete
2- plagiarism is too high and needs to be revised
3- What was the exact time of the study(how long?)
describe each drug completely (how much, how often and how long?)
4 - Discussion also is too too long, must be shortened to be more concrete
5- Methods in line 648 : authors wrote
Serum osteocalcin bone biomechanical strength, serum CTX, oxidative stress enzymes, gene expressions, and structural and static bone histomorphometry parameters
were all examined.
Authors must give details, code numbers, suppliers of each kit
Principle and a citing a reference for each assay must be provided
6- Figure 5 : need to add a clear scale bar of the images
High power images are not good enough and authors should provide also low power images and show the pathological features on the images itself by arrows or symbols
7- The type of ELISA kits should be provided (direct, indirect, Sandwitch ELISA,...etc)
Author Response

(The authors gave the same response as above.)

Round 3
Reviewer 2 Report
Comments and Suggestions for Authors
The revised version of paper titled (Comparative Analysis of Pure Tocotrienol and Tocotrienol Mixture: Differential Effects of Palm and Annatto Tocotrienol on Bone Health in Glucocorticoid-Induced Osteoporosis) by Authors Elvy Suhana Mohd Ramli and his colleagues was partly improved
I find it mandatory to
1) Figure resolution must be improved, the current status is not good enough
2) add the explanation of significance letters (a,b,c...etc ) in each figure legends.
How can the reader understand the symbols?
Please avoid repetetion of the same difference by two symbols
use a: different to group 1
b: different to group 2,...etc
Thanks
Author Response
Dear reviewer,
On behalf of the authors, I am resubmitting the third revision of the manuscript entitled ‘Comparative Bone-Protective Effects of Tocotrienol Isomers from Palm and Annatto in Dexamethasone-Induced Osteoporotic Male Rats’ Thank you for your valuable suggestions and comments. We tried our best to answer and respond to comments. We hope they meet your expectation and we are happy to respond to further suggestions. The changes are highlighted in yellow.
Best regards
Elvy Suhana Mohd Ramli
Corresponding author

Round 4
Reviewer 2 Report
Comments and Suggestions for Authors Manuscript ID: ijms-3818621 Type Article Title: Comparative Analysis of Pure Tocotrienol and Tocotrienol Mixture: Differential Effects of Palm and Annatto Tocotrienol on Bone Health in Glucocorticoid-Induced Osteoporosis Authors: Elvy Suhana Mohd Ramli * , Fairus A Hmad , Nur Aqilah Kamaruddin , Kok-Yong Chin , Kok-Lun Pang , Ima Nirwana Soelaiman Section: Molecular Pathology, Diagnostics, and Therapeutics Special Issue: Molecular Studies of Bone Biology and Bone Tissue: 2nd Edition Was improved significantly and I am pleased to recommend it for publication in IJMS Thanks